# In-Situ Comparative Study of Eucalyptus, Basil, Cloves, Thyme, Pine Tree, and Tea Tree Essential Oil Biocide Efficacy

**DOI:** 10.3390/mps5030037

**Published:** 2022-04-27

**Authors:** Andrea Macchia, Hélène Aureli, Fernanda Prestileo, Federico Ortenzi, Shaila Sellathurai, Antonella Docci, Eleonora Cerafogli, Irene Angela Colasanti, Michela Ricca, Mauro Francesco La Russa

**Affiliations:** 1YOCOCU, Youth in Conservation of Cultural Heritage, Via T. Tasso 108, 00185 Rome, Italy; andrea.macchia@uniroma1.it (A.M.); sh.sellath@icloud.com (S.S.); cerafogli.1559339@studenti.uniroma1.it (E.C.); colasanti.1748988@studenti.uniroma1.it (I.A.C.); 2National Research Council of Italy, Institute of Atmospheric Sciences and Climate (CNR-ISAC), Via Fosso del Cavaliere 100, 00133 Rome, Italy; fernanda.prestileo@cnr.it; 3Department of Biology, University of Rome Tor Vergata, Via della Ricerca Scientifica snc, 00133 Rome, Italy; federico.ortenzi@alumni.uniroma2.eu; 4Archaeological Park of Ostia Antica, Via dei Romagnoli 717, 00119 Rome, Italy; antonella.docci@beniculturali.it; 5Department of Biology, Ecology and Earth Science DIBEST, University of Calabria, Via Pietro Bucci, 87036 Arcavacata di Rende, Italy; michela.ricca@unical.it (M.R.); mlarussa@unical.it (M.F.L.R.)

**Keywords:** eucalyptus, basil, cloves, thyme, pine tree and tea tree essential oils, biocides, green conservation, stone materials, mosaic

## Abstract

Bio-colonization is a dynamic and multiphasic process headed by microorganisms. Conventional treatments to process affected stone materials include chemical biocides, whose formulations are mainly composed of quaternary ammonium salts(QAs), reported to be toxic for human health, dangerous for the environment, and not biodegradable. Accordingly, novel green and eco-friendly products are a promising alternative to treat stone materials deteriorated by microorganism colonization. In this study, the efficacy of pure essential oils (EOs) and a mix of EOs was assessed in situ and compared to a conventional biocide based on QAs, and two commercially green products based on EOs, which were taken as references, through application on a mosaic located at the Archaeological Park of Ostia Antica (Rome). The EO biocide efficacy was analyzed by ultraviolet induced luminescence, spectro-colorimetry and bio-luminometry analyses while the possibility of their permanence on simulated substrate was studied by FTIR spectroscopy. It was observed by FTIR analysis, that EOs considered volatile can leave a residue after the application; typical fingerprint bands at about 2926, 1510, and 1455 cm^−1^ were recorded in the EO spectra. Every tested oil was confirmed to have a biocide action although minimal in relation to the most conventional products based on QAs. The synergy of the essential oils revealed positive results, showing a stronger biocide efficacy. Further investigation should be carried out to develop the method of application and study of essential oils on cultural heritage.

## 1. Introduction

Bio-colonization is one of the main problems affecting the cultural heritage sector [1]. This phenomenon takes place on structures exposed to an environment characterized by specific local conditions, such as high moisture, high salinity, and abundance of organic nutrients [2,3]. According to the colonizing organisms and the characteristics of the interested area (surfaces, cracks, or pores), the entity of the damage changes, thus varying from exterior damage to irreversible disintegration of the inner substrate [4].

Therefore, research on the conservation of cultural heritage, as one of the main objectives, has to explore novel solutions aimed at removing and controlling the so-called biodeteriogens [5,6,7,8].

Currently, the strategy to contrast biodeterioration is based on the application of synthetic biocidal products, followed by a cleaning procedure to mechanically remove the remnants of the biofilms to prevent and/or hinder biodeterioration. Traditionally, the biocides derive from chemical products including acids, pyridines, quaternary ammonium salts, and organometallic compounds, with benzalkonium chloride, permethrin, and sodium fluoride [1] that are dangerous for human health and for the environment [9].

Modern commercial products were developed from quaternary ammonium compounds (QACs): Algophase and Preventol^®^RI50. Despite all of them possessing moderate toxicity, they show short-time effectiveness. Furthermore, QACs may be harmful not only for the operator but even for the environment [10,11,12]. Moreover, several compounds are not biodegradable and may cause uncontrollable contamination near the area of application [13]. In addition, Tabata and colleagues [14] reported *Pseudomonas species* to be responsible for the degradation of chloride compounds, contained in QACs, since microorganism consortia can feed on carbon sources from organic residuals contained in QACs, thus encouraging recolonization. Another issue occurs in repetitive QAC treatments, the onset of resistance of treated biocolonizers. This was apparent in the reported case study, in the Cave of Lascaux (France), where the fungus *Fusarium solani* during repeated treatments gained resistance, hence favoring the spread of other microorganisms (e.g., *Ochroconis lascauxensis*, *Ralstonia* spp. and *Pseudomonas* spp.) [15].

As a consequence, some of the most effective biocidal products have been recently banned due to their toxicity, with the result that a biocide that is effective and completely safe has not yet been developed. During the last decades the need to find environmental and human eco-friendly products has increased [16,17,18,19,20] with the focus of recent studies on the promising features of natural biocides, such as plant extracts or essential oils (EOs).

EOs are a complex mixture of organic volatile compounds that can be obtained from plants by hydro- or steam distillation [16]. Mainly characterized by terpenes, EOs include hydrocarbons, alcohols, ethers, aldehydes, ketones, and esters [21]. EO configuration depends on many factors including the genotype (species, cultivation and ecotype), the ecological factors (geographic origin, climate, and soil characteristics), and processing techniques [22]. These factors are responsible for the flavor and fragrance of aromatic plants and the antimicrobial activity [23]. Due to their composition, EOs possess a wide and diversified range of properties and are exploited in numerous fields including traditional medicine (used as an alternative to the conventional antibiotics), food (to protect and control rancidity), cosmetics (having antioxidant effects), and pharmaceuticals [22,23,24,25,26,27,28,29]. They produce a wide range of secondary metabolites with antibacterial, antioxidant, antimicrobial, antiviral, and anti-inflammatory properties.

Several researches have demonstrated that EOs efficiently contrast the growth of diverse strains of fungi and bacteria. EO activity in affecting microbial growth takes place through different pathways: indeed, EOs inhibit growth by affecting cytoplasmic membrane integrity, influencing cellular metabolism, affecting enzymatic activity, and impacting protein synthesis [13,30,31,32].

The biocidal effect of many EOs has already been tested, such as for: *Origanum vulgare*, *Rosmarinus officinalis*, *Lavandula angustifolia*, *Thymus vulgaris*, *Allium sativum*, *Pimpinella anisum*, *Eugenia caryophyllata*, *Calamintha nepeta*, *Cinnamum zeylanicum*, *Carum copticum*, *S. aromaticum*, *Citrus sinensis*, *Melaleuca alternifola*, *Cuminum cyminum*, *Eucalyptus globulus* etc. [1,16,29,33,34,35,36].

EOs oils obtained fromthyme and oregano plants have been reported to possess the most effective biocidal activity. Indeed, these EOs are mainly characterized by phenolic compounds, respectively thymol and carvacrol [30]. Thymol and carvacrol are monoterpenoid phenols that exhibit strong antimicrobial, anti-inflammatory, and antioxidant properties; another phenolic compound that shows these properties is eugenol, present in clove oil. On the contrary, EOs characterized by alcoholic compounds, such as tea tree oil (terpinen-4-ol and α-terpineol) express a slightly less effective action against microorganisms, showing antiseptic, cytotoxic, antifungal, antiviral and anti-inflammatory action and mainly used in the pharmaceutical and cosmetic industry [16,22]. Geranium oil (geraniol and citronellol), peppermint oil (menthol), and lavender oil (linalool and linalooll acetate) are members of this group. Rakotonirainy et al. [35], demonstrated the high antifungal activity of linalool and linalool acetate, present in *Lavandula angustifolia* against fungal strains isolated from library and archive storage areas. Ketones are the main constituents of sage oil (thujone, camphor) and peppermint oils (menthone, carvone), while fennel, eucalyptus and rosemary oils are rich in ethers: an ethole and 1,8-cineole, respectively. Rosemary extract, moreover, contains carnosic acid, as carnosol and rosmarinic which present antimicrobial, antioxidant, and anti-inflammatory properties.

Stupar and colleagues (2014) [37], applying the biocide activity of *Origanum vulgare*, *Rosmarinus officinalis*, and *Lavandula angustifolia* EOs, against fungal strains isolated from different artifacts, discovered that *O. vulgare* showed the strongest inhibitory response. Veneranda et al. [1] analyzed 10 different EO constituents that ratified thymol (*Thyme* sp. EO), eugenol (*Clove* sp. EO) and cinnamaldehyde (*Cinnamon* sp. EO) as the best enduring inhibitors.

However, reports about the application of EOs in cultural heritage conservation, as a valid alternative to traditional biocides, are still very limited to in vitro applications and need an application “on-field” to test the effectiveness, the proper way of application, and a monitoring strategy to define the efficiency over time. In vitro analysis is limited as biofilms are known to have different biological properties on stone surfaces than in suspension [10]. In addition, other factors such as the surface of application, the environment, and the weather could influence the effectiveness of essential oils in the field.

The aim of this study was to test a combination of treatments, especially focusing on evaluating the effectiveness of selected EOs according to their biocidal activity as defined by the literature, through the application in a real case study [1,38,39,40].

Pure EOs and mix of EOs were compared to a conventional product based on QACs and two commercially green products based on EOs, which were taken as references. The monitoring of their effectiveness was assessed when applied on a white and black mosaic in the XIX room of “Insula delle Muse” in the Archaeological Park of Ostia Antica (Rome).

## 2. Materials and Methods

Six EOs and their mixture were used as biocides and compared to three commercial products in the experimentation as reported in Table 1: Preventol^®^ RI50, based on QACs, and two commercially green products, Biotersus and Essenzio. Pure commercial EOs such as eucalyptus, basil, cloves, thyme, pine tree, and tea tree were tested and applied in June 2021 on the biodeteriorated mosaic surface by brushing on their prepared solutions. All the species were diluted with distilled water at a final concentration of 0.4% *v*/*v* as defined in the previously exhibited scientific literature [41]. The reported concentration used, by Amri and colleagues, was 0.4% *v*/*v* for *Pinus pinea* [39], while lower concentrations have been reported in literature for the other EOs. The oils were applied with the addition of 8.5 g of an agar thickener (NEVEK^®^ by CTS) to limit the evaporation of oil in the outdoor atmospheric conditions. To improve the homogeneity of water and oil dispersions, Tween^®^20 detergent was added at 0.3% *v*/*v* thus generating an emulsion.

An innovative green product (BioTersus-Exentiae), a mixture of EOs (*Cinnamonum zeylanicum* (0.25% *v*/*v*), *Eugenia caryophyllata* (0.5% *v*/*v*), *Corydo thymus capitatus* (0.4% *v*/*v*), and tensioactive Tween20 (0.3% *v*/*v*) was applied diluted at 1.3% with distilled water, by brushing it on the surface, as provided by the manufacturer [42].

A new product (YOCOCOIL) was tested, formulated and proposed by YOCOCU APS, a mix based on the combination of the most promising oils in the literature, *Eucalyptus globulus* (0.25% *v*/*v*), *Thymus vulgaris* (0.4% *v*/*v*), *Eugenia caryophyllus* (0.5% *v*/*v*), and *Ocimum basilicum* (0.25% *v*/*v*). For the formulation of this product, the commercial product Biotersus, also composed of EOs, was taken as reference. The mix was applied adding to the pure essential oils 8.5 g of NEVEK^®^ and Tween^®^20 at 0.3% *v*/*v* and diluted with distilled water.

As a reference for the current treatments, a traditional biocide was used, Preventol^®^ RI50, containing quaternary ammonium salts in diluted concentration 3% *v*/*v* in distilled water. The product was applied with a soft brush.

Finally, the experimentation considered a natural product Essenzio, an extract of *Thymus vulgaris* and *Origanum vulgare* that was employed by spraying the pure product on the surface, as reported in the data sheet [43].

The products were applied on a white and black mosaic in the XIX room of “Insula delle Muse”, in the Archaeological Park of Ostia Antica (Rome). The biofilm layer on the mosaic (Figure 1), observed under an optical microscope, produced by cyanobacteria, chlorophyta (photosynthetic prokaryotes), and green algae (*Chlorella*) represented pioneer phototrophic microorganisms on this kind of architecture [44].

Two portions of the mosaic were selected and delimited, respectively in Area 1 and Area 2, characterized by a different level of colonization as shown in Figure 2.

Area 1 presented by macroscopic and optical microscope observations, a minor entity of biodeterioration compared to Area 2, subject to a constant flux of water coming from the descending roof (Figure 2). Each of those two areas was subdivided into other ten zones respectively called P, A, B, C, D, E, F, G, H, and I, where the different EOs or products were applied, as reported in Table 1.

After the product application, to enable the EOs to operate on the biodeteriogens, the areas were covered with transparent film for three days, as the EOs are considered highly volatile and alsoto prevent their oxidation and to limit environmental influences on the tested area. To be effective and kill the organisms involved, the tested agents need to penetrate into the target cells of the complex biofilms composed of different phototrophic and heterotrophic microbes (such as fungi, algae, cyanobacteria, and bacteria) and extracellular polymeric substances (EPS) [1,10]. After the above-mentioned time the protective film was removed. A week later from the application of the biocide, the two target portions were slightly brushed, then the biocidal treatment was repeated to strengthen the effect. Finally, after the second biocide treatment, the areas were cleaned mechanically with soft brushes and washed with distilled water to remove totally the biological patina. Several analytical investigations, such as ultraviolet induced luminescence (UVL), spectro colorimetric analysis, and adenosine triphosphate (ATP) test, were used to define the surface state before the treatment and after the second biocide application.

The effectiveness of the treatments was investigated by measuring the chlorophyll fluorescence, the measuring of the presence of ATP nucleotide, and through the definition of the colorimetric parameters of the mosaic surface.

Each experiment was repeated on average 3 times. Data are presented as means ± SD. Statistical evaluation was conducted by Student’s *t*-test or a one-way ANOVA, followed by Tukey’s multiple comparison test (homogeneous variances). Statistical significance was at *p* < 0.05.

### 2.1. FTIR Analysis

As a preliminary test, Fourier transform infrared spectroscopy (FTIR) was adopted to analyze the EO residues after drying them with a heater at 40 °C for 32 h to accelerate evaporation. The time of drying was selected on the basis of the weather conditions related to the real application on the mosaic (end of June in Rome). The EOs were applied on a glass specimen slide and analyzed after evaporation to define the possibility of permanence on the substrate. The aim of this activity was to define the presence of residue due to oil application and to limit secondary chimism or chromatic alterations in the real mosaic [1,45]. FTIR spectroscopy was applied in attenuated reflectance modality (ATR). The IR spectra were collected employing the spectrometer Nicolet Summit FTIR equipped with the accessory Everest Diamond ATR, which allows the analysis in total attenuated reflectance (ATR). This technique allows the direct examination of the liquid and solid samples without preparation of the materials analyzed. The resolution of the instrument is 8 cm^−1^, the scans performed by the instrument on each sample were 32.

### 2.2. Ultraviolet Induced Luminescence (UVL)

Several pictures of the flooring were taken with visible light in order to evaluate the status of the mosaic, the extension of the biological patina, and the effectiveness of the biocides, before and after the treatments. Ultraviolet fluorescence induced by two Madatec spotlights at wavelengths of 365 nm (UV), was used to capture visible light-emitted by chlorophyll microorganisms still present on the mosaic surface after the cleaning procedures. Pictures of each area were taken before and after the biocide treatments using a Madatec multispectral system with the support of a Samsung NX50028.2 MPBSICMOS camera. To observe the fluorescence induced in the visible region by ultraviolet radiation, two filters were applied: HOYA UV-IR filter cut 52 and Yellow49552mm F-PROMRC022.

### 2.3. Spectro-Colorimetric Analysis

Spectro-colorimetric measurements, for each delimited area, were taken before and after the treatments in order to evaluate the chromatic variation of the parameters L*, a* and b* and to understand the surface color distance between the treated surface and untreated surface (with biological patina).

Spectro-colorimetric analysis (400–700 nm) was carried out by a portable spectrophotometer Y3060 (by 3nh), D65 illuminant, measurement aperture 8 mm, component SCI included. Data were analyzed using the CIELab color system.

The chromatic parameter a* corresponds to the change in redness–greenness of the surface analyzed, while b* is associated with the variation in yellowness–blueness and L represents the lightness–darkness. For each zone ΔL*, Δa*, Δb* and ΔE* were calculated taking into account the parameters L*, a*, and b* before the treatment with EOs and after the biocide procedure using the following formulas:Δa* = a*_2_ − a*_1_(1)
Δb* = b*_2_ − b*_1_
(2)
ΔL* = L_2_ − L_1_
(3)
ΔE* = [(ΔL*)^2^ + (Δa*)^2^ + (Δb*)^2^]½(4)
where L*_1_, a*_1_, and b*_1_ correspond to the measures before the application of the biocide and L*_2_, a*_2_, and b*_2_ after the second treatment. If Δa* results positive there is a variation of the surface analyzed versus red, vice-versa versus green; positive values of Δb* indicate yellowing, instead of negative blueing. An increase of ΔL value corresponds to a brighter area vice-versa to a darker facade.

A recolonization process can be observed when the values of ΔL* and Δa* decrease and Δb* increase indicating the surface analyzed is becoming darker and more green-yellowish due to the presence of phototrophic microorganisms [46]. Positive values of ΔL* and Δa* with a decrease of Δb*represent a decolonization process and therefore the successful biocidal action of the treatments. The total color change ΔE* gives information about the distance between two different points in the colorimetric space, for this research the color surface before and after the treatments. It indicates the total difference of the treated surface and how strong was the biocide action for each product tested.

### 2.4. Adenosine Triphosphate (ATP) Test

An average of three bio-luminometric measurements of adenosine triphosphate (ATP) were taken dabbing a 10 cm^2^ surface using a portable bio-luminometer, Lumitester PD-30 (Kikkoman) and LuciPacPen AQUA, for each area (1 and 2). An external area was analyzed and taken as a reference of anuncontaminated zone. The ATP nucleotide (adenosinetri-phosphate) is an energetic molecule contained in every cellular type. The instrument exploits the chemiluminescence reaction of the reagent luciferase, to emit light, in contact with ATP. The count of ATP detected from the surface, expressed in relative light units (RLU), reveals the metabolic activity of microorganisms or organic residues still present after the cleaning procedures.

## 3. Results and Discussions

### 3.1. Analysis of Residue Using FTIR Analysis

A 10 μL of each EO was applied on a glass slide for the analysis of FTIR in ATR spectroscopy, before and after being degraded for 32 h at about 40 °C in a heater [1].

The collected spectra reported a high variability of peaks due to multi-molecules that usually characterize the oils as terpenoids and derived hydrocarbons. In all the spectra EO residue was observed on the glass surface. Table 2 reports the main peaks identified in the spectra of the tested products in situ based on essential oils with the presence of the thickener, both before and after the drying process, and moreover the pure essential oils without the thickener analyzed after the drying process. The peaks detected at around 926, 911, and 763 cm^−1^ are related to the glass specimen slide. Peaks at around 3278, 1636, 1373, 1150, 1069, 1043, 966, 932, 713, and 687 cm^−1^ instead correspond to the thickener NEVEK. Peaks measured below 3064 (cis C=CH stretching), 2923 (CH2 asymmetric bending), 1744 (carbonyl stretching), 1635–1650 (RHC=CH2), 1511, 1450 and 1380 (CH2 bending), 1117 and 1097 cm^−1^ (C–O ether bending), 920–850 (isopropyl group), 810 (CH bending) can be considered the oil fingerprints [22,45]. To understand completely the obtained results (residues) of the tested reagents, an FTIR analysis was carried out using directly pure EOs, without the thickener. Additionally, in this case the spectra showed the presence of residues on the surface. The stability of EOs depends on several factors such as light, temperature, and oxygen. Around the temperature the stability decreases with a prolonged exposure to a value higher than 38 °C while degradation of terpenoids could be induced by temperatures above 100 °C [47]. The presence of thickener limited the evaporation of volatile fractions, increasing the stability and the time of application. Typical bands for cloves are shown at around 1462, 1510–1512(CH2 bending), 1591–1618 and 1255–1265 cm^−1^ (stretching vibration of C–O group). Thyme shows bands at 2956, 1255–1301 (CO stretching vibration), 1619, 1583, 1457 (CH2 stretching), and 808 (CH bending) and 738 cm^−1^. Pine tree spectra show common bands at 1622, 1446 (CH2 bending), 1379, and 936 cm^−1^. The common basil bands in all spectra are at 1462 (CH2 bending) and 2963. Bands at about 2850, 2915, 2942, 3006, 3034, 3086, 1470, 1510, 1610, 1639, 760, 822, 914, 994, 1042, 1102, 1174, 1246, 1290, and 1442 cm^−1^ correspond to estragole, one of the main compound present in basil EO [45]. Eucalyptus spectra show common bands at 2927 and 1448 cm^−1^ (CH2 group) meanwhile the tea tree at 3340 cm^−1^. Biotersus spectra show these common bands 2920 (CH2 asymmetric bending), 1452 (CH2 bending), 2047, 1255 (C–O stretching vibration), 814 (CH), and 658 cm^−1^ while YOCOCOIL shows bands at about 2924 (CH2 stretching vibration), 1511 and 1454 (CH2 bending), 2962, 2048, and 1267 cm^−1^ (C–O stretching vibration).

In Figure 3 it is possible to observe thespectra difference between the residues detected by the solution of essential oils applied in situ and the pure EOs. The main bands of the tested products are related to the glass specimen slide and thickener, however a few bands with low intensity can be linked to some of the residues of the oils. This is detected especially in the spectra of thyme that shows bands at 2959, 2926, and 1457 cm^−1^, pine tree at 2929, 1621, 1446, 1379, and 936, and basil at 1511 and 1455 cm^−1^ (Figure 3).

### 3.2. Ultraviolet Induced Luminescence (UVL)

In visible reflectance imaging (Table 3), photographs were taken before, after the treatment, and after the cleaning procedure. The zones that looked more colonized by the green patina were in the order F, G, E, B, D, C, A, P, H, and I for Area 1. As for Area 2 the zones more infested were in the order H, D, E, C, A, B, F, I, P, and G. After the treatments the intensity of color biofilm decreased especially, for Area 1 in zone P followed by I, H, G, A, F, E, B, C, and D. Meanwhile, for Area 2, it decreased in this order P, H, I, G, F, E, A, B, and C. The best biocide efficacy both in Area 1 and Area 2 after the cleaning procedure was obtained by YOCOCOIL (applied in zone E) and Preventol^®^ RI50 (zone P) followed by the products Biotersus and thyme (zone F and G) for Area 1 and thyme and pine tree (zone G and H) for Area 2. A lower biocide efficacy was observed for Area 1 in comparison with Area 2. Macroscopically, the patina changed color from green to grey with the biocide treatments. After the cleaning process, the patina on the surface was totally removed.

In Area 1, it was possible to distinguish a strong remitted light induced by chlorophyll molecules present in the microorganisms, particularly evident in zone F (Table 4). UV light confirmed the zones that seemed to have been more attacked by biodeteriogens in visible light, with a particular strong fluorescence being observed in the F, E, and G zones of Area 1. This fluorescence disappeared after the biocide application, as a result of the devitalization of the microorganisms. Such an effect can be observed especially in the E, C, and H zones of Area 1 where the product YOCOCOIL, Essenzio, and pine oil were applied, followed by a medium efficacy reported for the zones G, F, B, D, and P and a low action on I, B, and A.

In Area 2, the remitted light, before the treatment, was very reduced. After the biocide treatment the zones, except for the zone G, P, and E, were characterized by almost the same red-light intensity. For this Area the strongest biocide action was obtained certainly by Preventol^®^ RI50 and thyme oil, respectively in the region G and P. A moderate efficacy was achieved in order by the products applied in zone A, B, F, and E. Meanwhile Essenzio, pine tree, tea tree and clove oil showed a low antimicrobial activity. The best results among the oils were then detected by thyme oil for both Area 1 and Area 2 and pine tree for Area 1 as reported in Table 5.

### 3.3. Spectro Colorimetric Analysis

Figure 4 reports the variation of the colorimetric parameters (L*, a*, b*) and the total color change ΔE* of the treated zones with the selected biocides, in Area 1 and Area 2, calculated as the difference between the surface after the second biocide treatment and the untreated surface.

The average values of the chromatic parameters L*, a*, b*, before the treatment, in Area 1, were L* 36.81, a* −0.89, and b* 16.71 while in Area 2, the average values correspond to L* 44.41, a* −0.28, and b* 15.07. These data are taken as a reference to see if the variation of the parameters increased or decreased after the biocide treatment and the mechanical cleaning of the surface.

The chromatic parameter, which changed the most for all the reagents used in both the areas, is the lightness L*, as the removal of the greenish biological patina enabled re-emergence of the white color of the mosaic tesserae [48]. For Area 1, ΔL* results in being positive almost for every zone. The two negative values reported for A (−9,18) and I (−12,26), treated respectively with eucalyptus and tea tree oil are not taken into account as related to measures recorded on several grey tesserae of the mosaic. The value of Δa* resulted positive for every zone while Δb* gave mainly negative values, except for zone G, treated with thyme oil, where Δb* is positive (3.02). Compared to Area 1, Area 2 is characterized by smaller chromatic variations. ΔL* and Δa* show positive values for every measure, while Δb* results in being negative. For both the areas, the best results are obtained by the application of YOCOCOIL in zone E, followed by Essenzio in zone C of Area 2, as the chromatic parameter L* increases the most, which indicates a lighting trend of the surface according to the cleaning procedure, while Δa* and Δb* remain almost unchanged. These results were confirmed by the highest values of ΔE* reached by YOCOCOIL in zone E both of Area 1 (30,34) and Area 2 (26,14), and therefore the highest difference of the color surface detected before and after the application of the treatments. A positive trend was observed by the application of Essenzio and Biotersus in Area 2.

Figure 5 reports the full spectrum color measurement of the treated zones in comparison with the untreated zone’s color (plotted using a black curve). In all the spectra of the treated zones, the peak of chlorophyll absorption (near 670 nm) decreased and disappeared in Areas 1 and 2 only for the treatment with Preventol^®^ RI50. For the oils, the best results were obtained by I = Tea tree in both the areas and F = Biotersus, G = Thyme for Area 1. The spectra confirmed a higher efficacy of biocide treatments for Area 1 with respect to Area 2.

### 3.4. Adenosine Triphosphate (ATP) Test

Table 6 reports the bioluminometric measures of adenosine triphosphate (ATP) before (14 June 2021) and after (24 June 2021) the application of the biocides for Area 1 and Area 2 and of a zone not treated. The best treatments seem to be reached by YOCOCOIL and Preventol^®^ RI50 respectively in zone E and P for Area 1 and Area 2, showing the lowest residue of microorganism metabolic activity, although good results were obtained for thyme, Biotersus, and tea tree oil. Taking into account the major difference in the values of ATP before and after the treatments, this indicates the strongest biocide action for Area 1 was instead reached in zone P where Preventol^®^ RI50 was applied followed by YOCOCOIL in zone E and Biotersus in zone F. As for Area 2 the best results considering the variation of ATP were achieved by Preventol^®^ RI50, basil, and Biotersus respectively in zone P, B, and F.

Figure 6 shows the values of ATP measured before and after the application of the biocides in each treated portion of Area 1 and Area 2. The best results with the highest difference and the lowest values of ATP were recorded both for Area 1 and Area 2 in zone P where Preventol^®^ RI50 was applied followed by zone E treated with YOCOCOIL in Area 1 in line with the previous findings.

Table 7 are summarizes all the best results obtained from the different analytical analyses carried out on the treated surface of the mosaic. The best performances were achieved by Preventol^®^ RI50 in zone P for both Area 1 and Area 2, followed by YOCOCOIL in zone E and Biotersus in zone F with a minor efficacy. Amongst the oils, tea tree oil applied in zone I stands out.

## 4. Conclusions

In this study, on the base of scientific literature, the biocidal action of seven different products, mainly EOs, based on eucalyptus, basil, cloves, thyme, pine tree and tea tree oils and their mixture were tested in comparison with the traditional biocide Preventol^®^ RI50 and two commercial bio-nature products, Essenzio and Biotersus. The experimental research was carried out on a roman biodegraded mosaic located in the XIX room of “Insula delle Muse” at the Archaeological Park of Ostia Antica. The aim was to acquire data on the effectiveness of several EOs and natural products, applied in situ, in relation to one of the most diffuse biocides of which its effective antimicrobial action is more than known. Currently there is in fact a lack of information about their action on real case studies. The effectiveness of promising natural derived biocides in a real application was analyzed, using the minimum biocide efficacy concentrations defined in the scientific literature and, for the commercial products using the data reported in the technical sheet of the customer.

The mosaic was treated twice with the selected biocides and the patina was then completely removed. To determine the best essential oil treatment several analytical investigations applied directly in situ were used: UVL imaging based on the UV fluorescence of chlorophyll, spectro-colorimetry, and bio-luminometry by adenosine triphosphate test. Finally, to evaluate the possible residues of the treatments, FTIR spectroscopy in ATR was used. The obtained results were encouraging, confirming the efficacy of the EOs which had been mainly observed and studied in vitro and their minor biocide ability in comparison with the conventional biocide Preventol^®^ RI50. All tested products demonstrated a biocide efficacy however the synergy of different EOs seemed to have a stronger biocidal action compared to the single oils. Among the oils, the best results were obtained by using tea tree, pine tree and thyme. This data was confirmed by the major efficacy of YOCOCOIL with respect to Biotersus and Essenzio. YOCOCOIL is a mixture of *Pinuscembra* L., *Eucalyptus globulus*, *Thymus vulgaris*, *Eugenia caryophyllus*, and *Ocimum basilicum*. FTIR analysis showed the possibility of the presence of essential oil residues on the tested substrates, after simulating an ageing process and adding the thickener in the formulation of the biocide products directly using the pure EOs. These results, depending on several factors such as light, temperature, and oxygen, could be a warning for the application of essential oils, especially using a thickener. A deeper investigation focusing on this topic should be carried out with the view of obtaining an understanding of what the presence of EOs might imply.

At the moment, there is no uniform bibliography on how EOs should be applied and studied over time, hindering the understanding of the dynamics and effectiveness. This research intended to give a first insight on how the antimicrobial activity of EOs should be analyzed when applied in situ.

New research should be focused on creating a standard system of EO application procedure on different materials in situ on stonework of cultural heritage, on the analysis and monitoring of their effective action during time, and on their concentration/mixture and of the way of their use.

## Figures and Tables

**Figure 1 mps-05-00037-f001:**
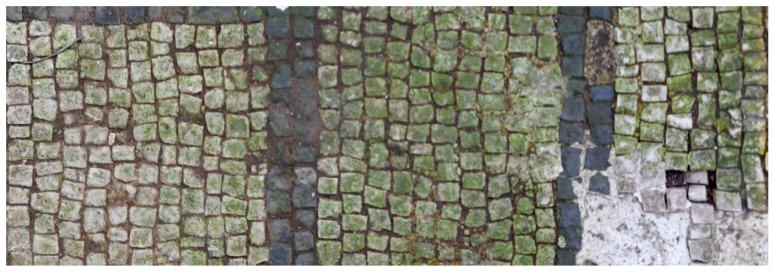
XIX room of “Insula delle Muse”: detail of the biofilm present on the mosaic.

**Figure 2 mps-05-00037-f002:**
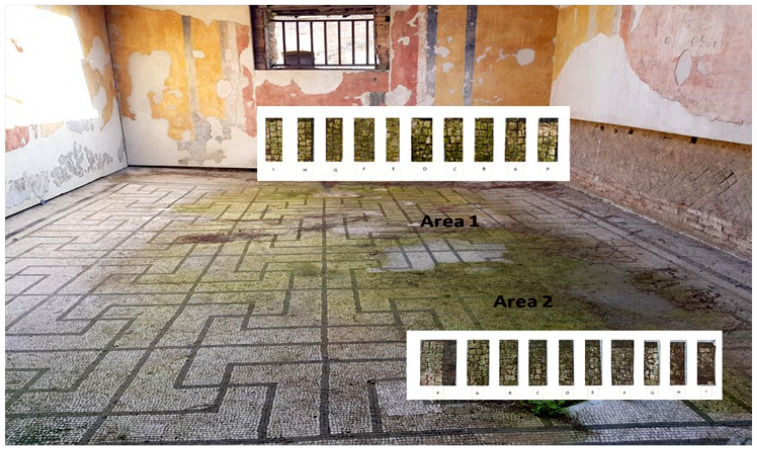
XIX room of “Insula delle Muse”: areas of experimentation with the indication of the ten tested EOS and products. Area 1 and Area 2 were each divided in zone P, A, B, C, D, E, F, G, H and I treated respectively with Preventol^®^ RI50, Eucaliptus, Basil, Essenzio, Cloves, YOCOCOIL, Biotersus, Thyme, Pine tree and Tea tree oil.

**Figure 3 mps-05-00037-f003:**
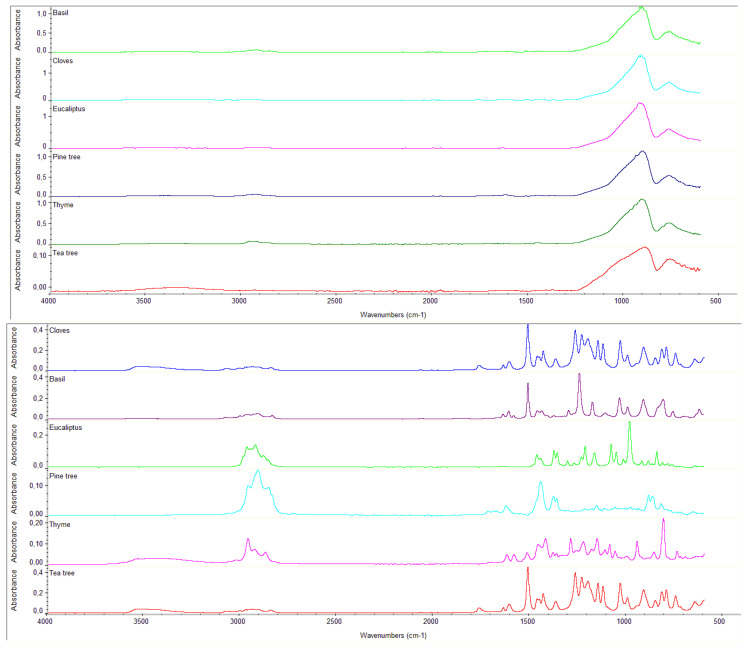
Overlaying of the FTIR spectra of the EOs tested without the presence of thickener and after drying (**top**) and with the presence of thickener before the drying process (**below**).

**Figure 4 mps-05-00037-f004:**
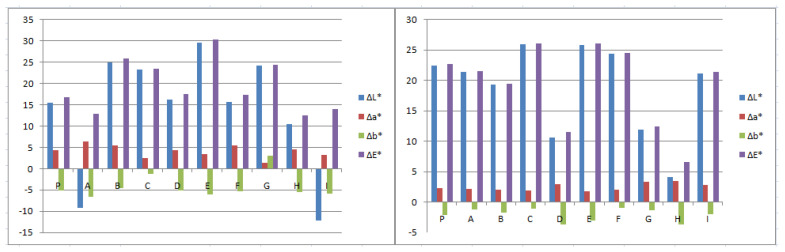
Histograms of the chromatic parameters (ΔL*, Δa*, Δb*, and ΔE*) for each treated zone of Area 1 (**left**) and Area 2 (**right**). Standard deviation is about 0.75 < SD > 2.90.

**Figure 5 mps-05-00037-f005:**
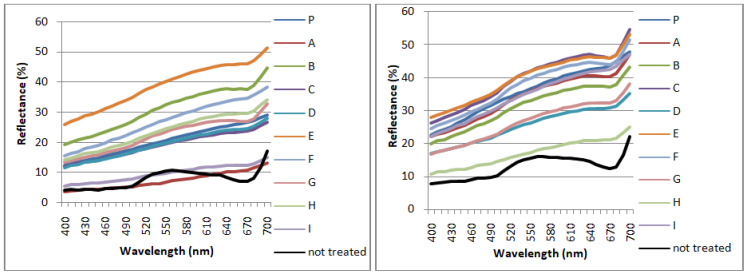
Full spectrum color measurement of treated zones in Area 1 (**left**) and Area 2 (**right**). P = Preventol^®^ RI50, A = Eucalyptus, B = Basil, C = Essenzio, D = Cloves, E = YOCOCOIL, F = Biotersus, G = Thyme, H = Pine tree, I = Tea tree and black = untreated zone. Standard deviation is about 0.75 < SD > 2.90.

**Figure 6 mps-05-00037-f006:**
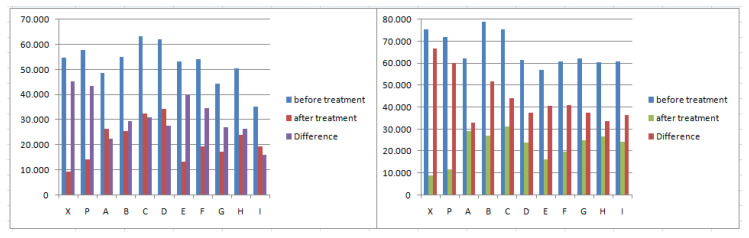
Histograms of the values of ATP measured before and after the treatment and their difference for each zone of Area 1 (**left**) and Area 2 (**right**).

**Table 1 mps-05-00037-t001:** Reagents used in the experimentation and zones of application.

Zones	Essential Oils	Composition	pH (Sd < 0.4)
A	*Eucalyptus globulus*	EO, NEVEK, Tween20	6.0
B	*Ocimum basilcum*	EO, NEVEK, Tween20	6.0
C	Essenzio	Extract of *Thymus vulgaris* and *Origanum vulgare*	4.6
D	*Eugenia caryophyllus*	EO, NEVEK, Tween20	6.0
E	YOCOCOIL	NEVEK, Tween20, *Pinus cembra* L., *Eucalyptus globulus*, *Thymus vulgaris*, *Eugenia caryophyllus*, *Ocimum basilicum*	6.4
F	Biotersus	*Cinnamonum zeylanicum*, *Eugenia caryophyllata*, *Corydothymus capitatus*, Tween20	4.5
G	*Thymus vulgaris*	EO, NEVEK, Tween20	6.5
H	*Pinus cembra* L.	EO, NEVEK, Tween20	6.5
I	*Melaleuca alternifolia*	EO, NEVEK, Tween20	6.5
P	Preventol^®^ RI50	Quaternary ammonium salts	6.7

**Table 2 mps-05-00037-t002:** FTIR-ATR main peaks related to the tested reagents: as pure compound without thickener (after the evaporation process), as applied products with thickener (before and after drying). The collected peaks related to the thickener NEVEK and to the glass specimen were removed from each measure present in the table.

Biocides	Bands	Bands	Bands
	Without Thickener (after Drying)	With Thickener (before Drying)	With Thickener (after Drying)
Eucalyptus	2928	2966-2943-2880-1464-1446-1359-1271-1233-1214-1166-1015-984-887-814-788-645	2927-1448-1370-1172-862
Basil	2925-1721-1511-1455-903	3077-3001-2963-2910-2834-1611-1584-1510-1463-1439-1300-1243-1176-1110-994-810-623	3417-2971-2439-2420-2322-2297-2182-2165-2108-2065-2022-1986-1917-1620-1462-1210-1174-1132-1091-946-859-820-806-788-742-699-683
Cloves	1765-1510	3521-3075-3003-2937-2842-1764-1606-1510-1462-1431-1265-1231-1199-1121-994-850-817-793-745-647	3425-2971-2896-2288-2190-2164-2050-1982-1457-1410-1263-876
YOCOCOIL	3454-2961-2919-2855- 2149-2048-2030-2002- 1976-1960-1735-1607-1511-1455-1351-1269-1249-1102-999-949-852-706-642-625	3348-2962-2924-1511-1449-1432-1267-1244-1214-1176-1121-989-840-807-741	3399-2971-2893-2325-2290-2189-2166-2105-2064-2050-1984-1563-1454-1263-1014-875-818-730-676
Biotersus	3338-2051	3399-2957-2924-2869-1734-1624-1513-1452-1429-1266-1234-1207-1178-1120-993-943-865-814-796-642	3364-2920-2322-2190-2166-2111-2077-2047-1983-1450-1354-1255-1019-888-818-658-631
Thyme	2959-2926-1457	3424-2960-2925-2869-1702-1619-1583-1519-1457-1419-1338-1289-1259-1224-1180-1112-1087-997-945-857-808-738	2960-2926-1457-1618-1583-1291-1257-809-740
Pine tree	3394-2929-1621-1446- 1379-936	2958-2909-2853-2725-1715-1688-1674-1624-1445-1377-1362-1300-1215-1182-1115-1022-1009-980-936-885-867-821-653	2921-1622-1446-1378-937
Tea tree	3340-889-864-799	3339-995-889-864-830-815-799-780	3340

**Table 3 mps-05-00037-t003:** Images of Areas 1 and 2 before, after the treatment with Eos, and after the cleaning procedure.

	Area 1	Area 2
Before the treatment	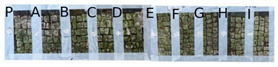	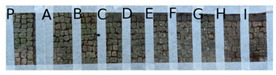
After the biocide treatment	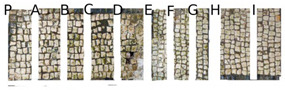	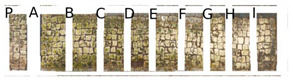
After the chemical and mechanical cleaning	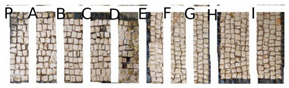	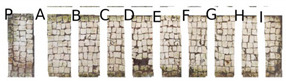

**Table 4 mps-05-00037-t004:** Images of areas 1 and 2 before the treatment with EOs and after the cleaning procedure in UVL.

	Before the Biocide Treatment	After the Biocide Treatment
Area 1	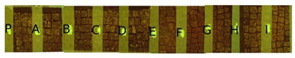	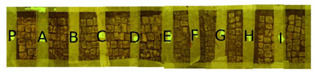
Area 2	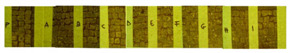	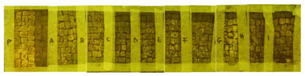

**Table 5 mps-05-00037-t005:** The level of efficacy of each product for Area 1 and 2.

Efficacy	P	A	B	C	D	E	F	G	H	I
Area 1	**	*	*	***	**	***	**	***	***	*
Area 2	***	***	**	*	*	**	**	***	*	**

* = low efficacy, ** = medium efficacy, *** = high efficacy.

**Table 6 mps-05-00037-t006:** Bioluminometric measures of ATP for each zone of Area 1 and Area 2.

	Biocides	Area 1				Area 2			
		Before	After	Difference	SD	Before	After	Difference	SD
X	Chemical and mechanical treatment	54,678	9327	45,351	<1102	75,346	8800	66,546	1310
P	Preventol^®^ RI50	57,617	14,261	43,356	<1772	71,971	11,896	60,075	1001
A	Eucalyptus	48,765	26,291	22,474	<1652	62,113	29,231	32,882	1641
B	Basil	54,891	25,447	29,444	<1586	78,830	27,001	51,829	896
C	Essenzio	63,375	32,542	30,833	<3206	75,344	31,244	44,100	1442
D	Cloves	61,932	34,341	27,591	<1188	61,344	23,789	37,555	1204
E	YOCOCOIL	53,168	13,351	39,817	<2325	56,793	16,101	40,692	2412
F	Biotersus	54,084	19,538	34,546	<2988	60,668	19,567	41,101	2191
G	Thyme	44,263	17,311	26,952	<692	62,231	24,878	37,353	1380
H	Pine tree	50,442	23,945	26,497	<1247	60,323	26,578	33,745	1567
I	Tea tree	35,261	19,321	15,940	<1281	60,641	24,080	36,561	2156

**Table 7 mps-05-00037-t007:** Best results for each analysis of the tested products for Area 1 and 2.

Biocides	Area 1	Area 2
Vis	UV	CIELab	ATP	Vis	UV	CIELab	ATP
Preventol^®^ RI50	x		x	xx	x	x	x	xx
Eucalyptus						x		
Basil								x
Essenzio		x					x	
Cloves								
YOCOCOIL	x	x	x	x	x		x	x
Biotersus			x				x	x
Thyme	x		x			x		
Pine tree		x			x			
Tea tree			x	x			x

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
