# Peer review of "In-Situ Comparative Study of Eucalyptus, Basil, Cloves, Thyme, Pine Tree, and Tea Tree Essential Oil Biocide Efficacy"

_mps, 2022, doi:10.3390/mps5030037_

Round 1

Reviewer 1 Report

The paper “In situ comparative study of Essential Oils biocides efficacy” by Macchia and coauthors deals with a very interesting topic, which was still scarcely discussed and investigated in conservation science, i.e. the use and performances of essential oils-based formulations as potentially effective green cleaning systems for the intervention on works of art where microbiological alteration patinae are present.

The authors nicely describe the conservative issue and the state of the art, presenting the motivation for their work in convincing way. However, unfortunately, the following doesn’t match the premises. More in detail, the methodologies and materials used in this work need some clarification, and – mainly – the results and discussion section needs a substantial strengthening, in order to support the conclusions in a solid manner.

I raise the following points, which could help the authors in improving their manuscript:

  1. The list of cleaning systems used is not so clear to me. In section 2, line 109 the author talk about 6 essential oils, while 10 different cleaning systems are reported in Table 1. Which was the rationale in choosing those cleaning systems? If I understand correctly, Preventol RI50 was selected as a reference biocide based on different chemicals (quaternary ammonium salts), while both Essenzio and BioTersus are two commercial products based on essential oils. Are they also included as references? And what about YOCOCOIL? Is this a registered trademark or a commercial product too? I could infer that it was something developed by the authors, just according to the affiliation of the first author of the present paper, but, in case, this could be more clearly stated somewhere.
  2. What about the concentrations of the different cleaning systems used? It is said that the 6 pure essential oils were diluted (in water, I guess) to 0.4 % v/v. Does this also applies to Essenzio, BioTersus and YOCOCOIL? Because it is not clear from the text, where (lines 118-124) higher concentrations seem to be reported for BioTersus and YOCOCOIL. This is obviously a crucial point, when comparing the effectiveness of different cleaning products. In other words, I would expect that the total concentration of the essential oils included in the YOCOCOIL system, for instance, would be 0.4%, as the one of the single essential oils used in the study.
  3. IF the YOCOCOIL system is indeed proposed by the authors (as I hypothesis – if I’m wrong, please, don’t consider this point) how did they come to this particular formulation? Did they do some preliminary testing? Could they comment on this?
  4. The application of the Essenzio product is completely different from that of all the other cleaning systems. This formulate is sprayed over the area, instead of being applied thickened with NEVEK. Why? Does this affect the comparison in some way? Can the authors please comment on this?
  5. The use of Tween20 is a bit unclear to me. Did the authors obtain emulsions? Microemulsions? Did they check the phase diagrams of the obtained ternary (or pseudo-ternary) systems made of essential oil(s), water and surfactant, in the absence of the thickener? Did they experiment some phase separation? Did they check for the phase stability of the systems over time? Do they think that a phase separation could affect the cleaning performances to some extent? I would like some more comments and thoughts on these aspects, which seem relevant to me.
  6. A common problem to the evaluation of cleaning results where some mechanical action is needed is that some objectivity is inherently lost with this cleaning methodology. Is there a way in this specific context, in order to avoid this inconvenience and estimate the effectiveness of the biocide systems tested before some mechanical action is used to remove the treated biological patina?
  7. I don’t understand exactly the cleaning procedure: the authors say that the cleaning systems were covered with plastic sheets (if I understand correctly) for three days, then the film is removed and only after a week the areas are mechanically cleared. Why? What happened in that 4 days gap?
  8. Section 3.1 needs to be clarified and strengthened. It is not clear which are the most important results. I understand that after 32 hours at 40 °C some residues are still present on the glass slides, but the authors do not state if this is a good result, or something to be concerned about. Moreover, they do not say anything about the chemical nature of the residues.
  9. The FTIR-ATR spectra need to be reported (at least in a SI file). Table 2 alone is not enough to understand clearly what happens.
  10. The pictures reported in Tables 3 and 4 are not clearly visible. It is almost impossible distinguishing the differences in the treated areas, and since there is no a numerical quantification in this case, it is important that the reader can actually see images of good quality and suitable size.
  11. The authors should clarify and expand the discussion reported in section 3.2. The work includes 10 different products, and not even all of them are cited in the text. The reader could benefit from a comparative table with a numerical score or some indication of the effectiveness of the different cleaning systems.
  12. Table 5 could be converted into a histogram. The reader could greatly benefit from some clarification. Also, why didn’t the authors calculate the total delta E? It could have given some useful information.
  13. Table 6 also should be improved. I think that the authors should at least include the difference between before and after measurements, which is the interesting observable. However, also in this case, a histogram would be more immediately understandable. Finally, what SD mean? Is it the standard deviation? And, in case, what does “< number” mean?
  14. The conclusions should be improved and rewritten according to the extensive modification required for the rest of the manuscript, paying particular attention to give convincing explanations about the most relevant results.

Furthermore, the text should be carefully checked for some minor issues relating to typos and small mistakes.

For the above considerations, I advise that the manuscript is accepted for publication only after the authors carefully addressed the aforementioned issues providing an extensively revised version.

Reviewer 2 Report

The overall manuscript needs critical English editing, however, the work is interesting and could be accepted after the revision regarding the following points:

Title:

Please name the six essential oils used in the study within the title of the paper.

Abstract:

The abstract is very short, I think, it should be elaborated to give details about the significance of the work and some of the key results as numbers, e.g., results for FT-IR and colorimetric assays.

Keywords:

Please mention the essentials oils used in the study within the keywords.

Introduction:

Line 61-62: please revise the sentence, replace “carbohydrates” with “hydrocarbons or terpenes”.

Line 62-63: also revise the sentence “These latter are responsible for the flavor, fragrance and antimicrobial activity.” as all types of volatile oils including hydrocarbons “terpenes” could be responsible for the odors of the aromatic plants.

Line 63-65: please also mention that volatile oils and plants rich in volatile oils are also consumed in several traditional uses including traditional medicine and cite the following reference: https://doi.org/10.3390/antiox11020332

Line 101: delete “then”

Methods:

Please insert a section for the statistical analysis.

Results and discussion:

Line 229: footprints or fingerprint

Section 3.1: figure for the FT-IR should be inserted, also more discussion is required to show the differences and similarities between the parameters.

Table 5: why there are no statistical variation notes, also in figure 3.

Reviewer 3 Report

The authors have chosen an interesting topic of study. However, the methods were unclearly presented and the manuscript suffered from careless presentation.

Minor comments below

Title
In situ in italics
Essential Oils -> essential oils

Abstract
20 ultraviolet-induced
20 Luminescence -> luminescence

Introduction
43 effectiveness, furthermore -> effectiveness. Furthermore,
46 Champanac.. Add reference number!
53 Remove "unreported before"
58 Why do you use capitals for EOs?
71-75 Latin names should be written in italics.
81 Tea tree -> tea tree
84 Remove " in 2005"
94 analyzing -> analyzed
101-103 Define the aim of the study more thoroughly.

M&M
111 by using a soft brush? Explain this more clearly.
117 .. -> .
121 Technical sheet? Explain, add a reference.
134 Remove Marasco et al. 2016
139-140 Present the areas more clearly.
153-155 Explain these methods closely.
185 2.3 Spectro-colorimetric analysis
191 ( by 3nh) -> (by 3nh)

208 Why 2.4 is bigger in size?
Adenosine TriPhosphate (ATP) test -> Adenosine triphosphate (ATP) test

Results
220 Why the first sentence is in separate chapter?
Tab. 2 -> Table 2.
You should refer to Table 2 in the text similar to the table heading.

References
356 ANderson -> Anderson
All the reference should be listed according to journal instructions.
Please check!

Round 2

Reviewer 1 Report

I read the reworked manuscript and the (honestly quite succint) replies, which the authors gave to the points I raised in the first round of review.

The addition of FTIR spectra, the reworking of some figures, the inclusion of histograms instead of hardly-readable tables, together with the strenghening of the results' discussion are changes that hopefully will help readers in understanding better the work. I appreciated the inclusion of a summarizing table before the conclusions - I found it very useful.

I still think that the pictures included in Tables 3 and 4 are completely useless in this form, as they are too small to be deciphered.

I believe, however, that the overall quality of the paper was improved, and that now it can be published as it is.

Reviewer 2 Report

Paper improved  

Reviewer 3 Report

Authors have made changes based on reviewers´ suggestions. I have no further comments.